# Injury Characteristics among Young Adults during and Immediately after the COVID-19 Lockdown

**DOI:** 10.3390/ijerph19158982

**Published:** 2022-07-23

**Authors:** Marcos Quintana-Cepedal, Miguel Ángel Rodríguez, Irene Crespo, Nicolás Terrados, Beatriz Sánchez Martínez, Miguel del Valle, Hugo Olmedillas

**Affiliations:** 1Department of Functional Biology, Universidad de Oviedo, 33006 Oviedo, Spain; marcosquintana99@gmail.com (M.Q.-C.); miguerguez95@gmail.com (M.Á.R.); crespoirene@uniovi.es (I.C.); 2Health Research Institute of the Principality of Asturias (ISPA), 33011 Oviedo, Spain; nterrados@aviles.es; 3Regional Unit of Sport Medicine-Avilés City-Council Foundation, 33401 Avilés, Spain; 4Department of Education Sciences, Universidad de Oviedo, 33005 Oviedo, Spain; bsanchez@uniovi.es; 5Department of Cellular Morphology and Biology, Universidad de Oviedo, 33006 Oviedo, Spain; miva@uniovi.es

**Keywords:** SARS CoV-2, physical fitness, recreational athlete, physical activity, musculoskeletal tissue, epidemiology

## Abstract

The lockdown due to the COVID-19 pandemic inherently changed people’s lifestyles. Forty-eight days of isolation led to worsening physical fitness in addition to the development of other unhealthy habits. The aim of this study was to describe sport-related injuries in the active general population. Physical therapy centres and sports medicine clinics were contacted via e-mail, seeking patients who had sustained an injury during or immediately (up to two weeks) after the lockdown. Patients who agreed to participate completed an online survey that followed the International Olympic Committee Statement. The questionnaire focused on physical exercise habits, type of injury, location and tissue affected. A total of 51 females and 67 males (30.5, SD = 8.8 years) participated in the study. Eighty percent of the participants performed aerobic training sessions, while the rest dedicated their workouts to strength training. Two in every three injuries were located in the lower limbs, and 80% affected the musculoskeletal tissue. Of all the injuries recorded, 67% occurred during the first week after lockdown. The number of aerobic exercise sessions was positively correlated with lower limb injuries (χ^2^ = 17.12, *p* < 0.05). Exercise habits should be considered when planning to return to a sport after a period of confinement to avoid injury.

## 1. Introduction

The COVID-19 pandemic has drastically changed people’s way of life. Lockdown was established as a measure to prevent the extreme transmission of the virus and protect the health of the population [1]. To date, approximately 400 million people worldwide are isolated in their homes due to a rise in infection rates [2,3], and potential lockdowns will be applied by governments in the future, leading to new pandemics within pandemics. In addition to the psychosocial consequences of home quarantines, including anxiety, depression and insomnia [4], physical fitness has also been widely compromised. Lockdown limited the opportunities to exercise and partake in physical activity since gyms were closed and access to outdoor environments was severely restricted. Prolonged home stays may result in increased sedentary behaviour—spending long periods of the day sitting or lying down [5]. Periods of detraining longer than four weeks have shown a marked decline in VO_2_max, while fewer impairments have been concluded in strength performance [6,7]. This situation leads to an impaired health status and might lead to worse chronic conditions [8]. In fact, there is a strong correlation between being physically inactive and adverse outcomes due to COVID-19, such as hospitalisation, admission to the intensive care unit and death [9]. Therefore, regular exercise, coupled with an optimal nutritional intake, is essential to preserve health [10].

To face the problem of physical inactivity, the experts made a series of recommendations to favour physical activity during lockdown and adapt it to the home setting [11]. However, the vast majority of studies examining changes in physical activity from before to during the lockdown caused by the COVID-19 pandemic revealed a drop in physical activity during that period [12]. According to a Greek population-based study, physical activity levels were drastically reduced by 16.3% during lockdown [13]. Among 1980 German students, 44.5% saw a decrease from pre-lockdown amounts of physical activity, while 32.8% increased the hours per week of training [14]. This same pattern was observed in Italian students, for whom physical activity was stratified by volume (high, moderate, low levels and inactivity); after one year of the pandemic, physical activity levels had dropped by 18% and 11% in high and moderate exercisers, respectively, while inactivity increased by 11% [15]. Children and adolescents, especially those living in urban environments, also showed an important decrease in physical activity levels [16]. Generally, individuals who were highly active pre-lockdown appeared to reduce their levels of physical activity during lockdown. A different pattern emerged from subjects who were moderately active in the pre-lockdown, who showed increased levels of physical activity during the period of home quarantine [17]. This circumstance could be interpreted as an opportunity generated by lockdowns for non-exercisers to begin a regular exercise routine [18]. An inverse association between the collateral effects of home confinement and the training volume prior to and during lockdown was concluded in Spanish recreational runners according to a study investigating the effects of a 48-day home quarantine on the first outdoor running session [19]. Considering the published data, intensity-related and weight-bearing exercises were the most practiced during lockdown (75%), with more than 60% of people training on their own [20]. Apart from the fact that this circumstance may increase the risk of injury, it is worth mentioning that home training routines present marked differences when compared to usual outdoor activity [21]. There are several sports or exercise modalities (e.g., running, swimming, skating and CrossFit^®^) whose movement patterns and skills cannot be executed and reproduced in the home setting for most people. The lack of specific training stimuli may result in the corresponding loss of neuromuscular adaptations [22]. In this regard, a progressive and programmed return to normal physical exercise is essential following lockdown to minimise the risk of injuries [23].

To the best of our knowledge, the literature on exercise-related injuries following home lockdown is scarce. Therefore, this study seeks to describe the effects caused by prolonged home confinement on the injuries sustained by a general population of Spanish adults while following the IOC Statement on Reporting of Epidemiology Data [24].

## 2. Materials and Methods

This study was observational and retrospective in design. It consisted of a self-reported electronic questionnaire created to be filled out by people injured during and immediately after the first COVID-19 lockdown (https://forms.gle/5snDTpGjnwZQvQyJ6, accessed on 20 July 2022) and aimed to record data on physical activity habits and injury characteristics. The questionnaire was structured in three sections. The first (1) collected information on age, gender and residential area. The second (2) collected physical activity characteristics (no. of sessions per week, type and duration of session, and physical activity habits before lockdown). The third (3) collected data on injury characteristics and previous injury history. Body region, type of injury and exercise exposure were recorded following the principles of the IOC Consensus Statement [24]. Physiotherapy centres and sports medicine clinics from around the country were contacted via e-mail between May and June 2020, seeking patients with exercise-related injuries. Participants answered the questionnaires after injury diagnosis. Participants who consented to participate were included if they had suffered an injury while performing physical exercise. People who were not able to understand the questionnaire were excluded. All participants gave their consent to participate in the study, and the principles of the Helsinki Declaration were followed.

An injury was defined as tissue damage or other derangement of normal physical function due to participation in sports resulting from rapid or repetitive transfer of kinetic energy [24]. Exercise exposure was calculated via the following formula (No. Weeks × H_ex_), where No. Weeks is the number of weeks that the study lasted and H_ex_ is the time (h) each participant dedicated to do physical exercise. The incidence was calculated for specific tissue affected via the following formula: ([Σ injuries/Σ exposure hours] × 1000) [25].

### Statistical Analyses

Descriptive variables were reported as frequencies and percent values or as mean and standard deviation when necessary. The Kolmogorov–Smirnov test was applied to assess the normality of the data. Pearson’s chi-squared test was applied to compare between type of physical activity performed and injury location. When appropriate, we have included the contingency coefficient (CC) to assess the strength of the relation. Statistical significance was set at *p* < 0.05. All statistical analyses were performed using the IBM SPSS Statistics V 27.0 statistical software (Chicago, IL, USA).

## 3. Results

Overall, 51 females and 67 males (30.5; SD = 8.8 years; *p* = 0.7) participated in the study. All the participants were healthy adults that performed physical activity noncompetitively and had not suffered COVID-19 before or during the study. The exercise characteristics are displayed in Table 1.

The total exercise exposure time was 3773 h, and the individual exposure was 28 h. Twenty-five (21.18%) participants were injured during the lockdown—79 (67%) in the first week after confinement and 14 (12%) in the second week. The distribution of injuries was diverse, with the lower limbs being the most injured area (66.1%). Frequencies for anatomical and tissue distribution are reported in Table 2. Regarding the tissue affected, most injuries were muscle/tendon (81.4%), ligament (6.8%), nervous (5.9%), bone (2.5%) and others (3.4%) [*p* < 0.05]. Injury incidence per tissue affected is shown in Table 3.

A significant association was found between the type and location of injury [χ^2^ (16) = 109.50, *p* < 0.05], showing a moderate correlation (CC = 0.69 *p* < 0.05). Aerobic training sessions were associated with the location and type of the injury [χ^2^ (8) = 17.12; χ^2^ (4) = 9.60; *p* < 0.05] mostly being in the lower limbs and muscles, respectively.

## 4. Discussion

The main findings from this study are a very high number of injuries during the first week after lockdown in a cohort of physically active Spanish adults. Two thirds of injuries affected the lower extremities, and 81.4% of injuries sustained were to the muscle tissue. A relationship was found between aerobic training sessions and lower limb injury.

The main health and exercise organisations recommend performing 150–300 min of moderate-intensity, 75–150 min of vigorous-intensity exercise, or some equivalent combination of moderate-intensity and vigorous-intensity aerobic physical activity per week [26,27]. In this regard, the participants in our study were physically active; two thirds of the surveyed population worked out more than 150 min per week, but 78% of the participants performed exclusively aerobic training during this period. Despite achieving the recommendations for aerobic exercise, the proposed standards in terms of strength exercise, balance and coordination were not met [11,26,27]. Most of our participants modified their exercise-related habits due to a lack of space and equipment. Exercise specificity is key to gaining tissue adaptations, but access to certain materials is a requirement for exercising correctly. A period of lockdown poses a challenge to exercisers, and it is further amplified for non-exercisers.

Our results agree with the general statement that compliance with physical activity recommendations allows for the maintenance of health and partially prevents the appearance of an injury. Malone et al. [28] observed that having a stronger lower body in terms of maximal and explosive strength was a protective proxy for developing injury in a sporting environment. Nevertheless, compliance with volume and intensity does not exempt the exerciser from extrinsic and intrinsic factors that may facilitate the appearance of a musculoskeletal injury. Physical fitness cannot explain the whole picture; it is necessary to remember that psychosocial factors may predispose individuals to an increased risk of injury as well [29,30]. Indeed, it has been observed that various behaviours of an emotional nature arose during lockdown, which could be associated with the high number of injuries observed [4]. The relationship between exercise and mental health is complex; exercise has been shown to be effective in improving mental health both in the general population and in those with physical or mental illness [31,32,33]. Therefore, exercise could have been implemented in our population with the aim of improving participants’ mental well-being and it, combined with higher levels of cortisol, could explain the results. Unfortunately, the collection of mental health disorders, such as anxiety or stress, was not part of the scope of this study.

Injury incidence significantly increased in athletes from professional leagues who suffered more injuries after lockdown despite team efforts [21,34,35]. This same pattern was observed in our participants, who were based on non-competitive individuals who lacked monitoring from sport science personnel. This similarity among heterogeneous sport populations could be explained by the reduced training load or the use of coadjutant exercises that had to be implemented in home environments, leading to worse muscular adaptations, so that when people could return to their regular physical activity, a misbelief that training during this period had been correctly implemented could have distorted real muscular fitness [6], resulting in an increment in the number of injuries. The COVID-19 pandemic seems to be a relevant factor with unknown magnitude acting in the complex interplay of sports, injury and performance. Injury databases can be used to analyse this interplay, but injury numbers must be interpreted with caution due to the many changes in exposure and type of sport-specific loading [36].

Of all the injuries recorded, 67% occurred during the first week after confinement. This might be linked to training load management errors, since peaks in load can lead to injury, especially in the amateur population [37]. Mosqueira-Ourens et al. [19] recorded the characteristics of the first outdoor running session after quarantine. They observed that advanced runners who covered longer distances during quarantine were capable of coping with the load in the first outdoor running session without pain. Runners who were trained at a higher level prior to or during confinement were likely better protected against injury risk when they returned to previous physical activity levels. It is possible that our population was not sufficiently prepared to tolerate the training loads, and this consequently resulted in a higher risk of injury. This might be explained by a lower chronic load and a rapid load spike after lockdown, which has been proposed as a risk factor [37]. Sixty-eight percent of our population trained more than 4 days a week, but this might not be stimulus enough to accomplish specific muscular adaptations. When exercising regularly for health, a series of positive adaptations appears that protects the body against oxidative damage. However, if the magnitude of exercise is too high, detrimental effects occur, which include mitochondrial dysfunction, inflammation or insulin resistance. Cell damage due to strenuous exercise may lead to tissue inadaptability and injury in consequence [38].

Another core finding from this research is the observation of two in every three injuries affecting the lower extremities. This is in accordance with other studies that examined the epidemiology of a variety of sports. In a study performed on collegiate athletes from various sports, Hootman et al. [39] observed that 53% of all recorded injuries were to the lower extremities. Similarly, athletes participating in the Olympics injure the knee, thigh, ankle and lower leg the most in all disciplines that are part of the Olympics [40]. The lower body seems to be more susceptible to injuries across a variety of sports and populations, and these observations agree with our findings. Considering injuries by tissue, 81% of our cohort sustained a muscle/tendon injury. Muscle/tendon injuries are among the most common in sport and require a mean of 15 days (SD = 17) to rehabilitate [25]. A period of insufficient training stimulus derives in muscular negative adaptations, fibre distribution, cross-sectional area, force production, power or isokinetic strength [7]. These mechanisms place individuals at a higher risk of sustaining an injury if the return to sporting activity is not managed properly [37].

Taking all the evidence together, injuries after a period of lockdown notably increase because of the interrelationship between various factors. Variations in nutrition, sleep and exercise habits are likely the cause of this pandemic of injuries. When returning to a habitual physical activity schedule, it is necessary to assess physical condition and ensure that the individual is healthy and ready to return to play.

This study is not without limitations. First, the study design was retrospective; participants were recruited by convenience sampling, and selection bias could have occurred. However, this is a more feasible form of collecting data that can help in the design of future prospective studies. General incidence could not be calculated since only injured people were surveyed and calculations without healthy individuals would have overestimated the result. Second, no distinction between the lower body regions was made. However, the least possible number of questions was asked to improve the response rate, and assessing every region would have resulted in a lower response rate. This is the first study to compare sports-related injuries while adhering to the IOC framework. However, we used a questionnaire that is typically applied in sporting environments, and reliability testing for our population was not performed. Therefore, results should be taken with caution [24].

## 5. Conclusions

In conclusion, the general population is at high risk of sustaining an exercise-related injury for several reasons, including the dosification of exercise modalities. Most of the injuries occurred during the first week after lockdown, and the lower limbs and the muscle tissue were at higher risk of sustaining an injury in these cases. It has been observed that performing aerobic training sessions exclusively is associated with lower limb injuries. The results of the current study may be used as the initial hypotheses to be studied in future prospective studies.

## Figures and Tables

**Table 1 ijerph-19-08982-t001:** Physical activity characteristics.

Variable		All (*n* = 118) %	Males (*n* = 67) %	Females (*n* = 51) %	χ^2^ (*p*)
Weight	No change	33.1	26.9	41.1	8 (0.02)
	Increase	32.2	35.8	27.5	
	Decrease	34.7	37.3	31.4	
Type of session	Aerobic	78	77.6	78.4	1.5 (0.46)
	Strength	22	22.4	21.6	
Physical Activity	<150 min/w	29.7	41.3	27.5	0.2 (0.65)
	>150 min/w	70.3	68.7	72.5	
Training Frequency	<3 d	31.4	32.8	29.4	2.9 (0.23)
	4–6 d	46.6	50.7	41.2	
	7 d	22	16.5	29.4	

**Table 2 ijerph-19-08982-t002:** Anatomical and tissue distribution of injuries.

Tissue	No. of Cases
Muscle/Tendon	96 (81%)
Ligament	8 (7%)
Nerve	7 (6%)
Bone	3 (2.5%)
Trunk	26 (22%)
Neck	1 (0.8%)
Upper Limb	7 (6%)
Lower Limb	78 (66%)

**Table 3 ijerph-19-08982-t003:** Injury incidence per tissue.

Tissue	Incidence
Muscle/Tendon	25.44 (22.8 to 28)
Ligament	2.12 (1.37 to 2.87)
Nerve	1.85 (1.15 to 2.55)
Bone	0.8 (0.33 to 1.25)
DK/NA	1.06 (0.53 to 1.6)

Incidence is reported as cases (CI 95%). DK/NA = Do Not Know, No Answer.

## Data Availability

Not applicable.

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
