# Peer review of "Injury Characteristics among Young Adults during and Immediately after the COVID-19 Lockdown"

_ijerph, 2022, doi:10.3390/ijerph19158982_

Round 1

Reviewer 1 Report

The COVID-19 pandemic is posing a very serious challenge to our societies as entire populations have been asked to restrict their social interactions and in many countries even to self-isolate and live in home-confinement for several weeks to months. This period of restricted movement affects all citizens regardless of age, sex and ethnicity. It forces people, even the youngest and fittest, to become suddenly inactive and adopt sedentary behaviours. Training cessation has been shown to negatively affect physical human performance, but very little is known about the effects of training stimuli reduction on the incidence sport injuries. Calculating injury risk estimates and defining risk profiles regarding types of training and exercise load will give more insight. This is why current study produces new knowledge instead of summarizing what is already known.

There are however some issues that deserve mentioning.

MATERIALS AND METHODS

A retrospective study design utilizing is a relatively quick and inexpensive way to collect pilot data. This can be helpful in identifying feasibility issues and designing a future prospective study. A major weakness of such retrospective studies is that the participants are often recruited by convenience sampling and not representative of the general population (selection bias).  Furthermore there is lack of predetermined sample size. Most sources of error due to confounding and bias are more common in retrospective studies.

Please specify participant’s educational or occupational background, demographics and other quantifiable characteristics. Have some participants suffered from a COVID-19 infection?

The authors wrote:

 Page 3, line 102

….”was estimated by multiplying the number of weeks that the study lasted by the hours each participant dedicated to doing physical exercise.”

Better to write down the formula:  

number of weeks x (hours /week)  dedicated to doing physical exercise

Page 3, line 108

“Pearson´s Chi Square test was applied for comparison of variables.”

 Which variables were compared?

Please give a detailed description of how statistical analyses were performed.

RESULTS

Please report clearly the results of the statistical analysis

Every statistical test that you report should relate directly to a hypothesis (eg null hypothesis). Begin the results section by restating each hypothesis, then state whether your results supported it, then give the data and statistics that allowed you to draw this conclusion.

Please insert a table representing the different types of injury, versus the anatomical distribution (number [%])

DISCUSSION

The COVID-19 pandemic seems to be a relevant factor with unknown magnitude acting in the complex interplay of sports, injury and performance. Injury databases can be used for analysing this interplay, but injury numbers need to be interpreted with caution due to the many changes in exposure and type of sport specific loading. [1]

COVID-19 could be responsible for a decline of efficiency of neuromuscular system, changes of body mass and composition and a consequent loss in terms of performance and endurance with a higher risk of injury. With regard to the lower limbs, it would be interesting to know the incidence of knee injuries due to different types of training programs. Certain injuries may predispose a knee to accelerated osteoarthritis.  Until now there are no evidence-based strategies for the return to sport activity. [2]

Can the authors comment upon this.

The retrospective design should be mentioned as a limitation.

The results of current survey may be used as the initial study generating hypotheses to be studied further by larger prospective studies.

[1] Tak I, Rutten J, van Goeverden W, Barendrecht M. Sports participation and injury related to the COVID-19 pandemic: will data support observations from clinicians and athletes? BMJ Open Sport Exerc Med. 2022 Mar 1;8(1):e001317. doi: 10.1136/bmjsem-2022-001317. PMID: 35251691; PMCID: PMC8889451.

[2] Bisciotti, G.N.; Eirale, C.; Corsini, A.; Baudot, C.; Saillant, G.; Chalabi, H. Return to football training and competition after lockdown caused by the COVID-19 pandemic: Medical recommendations. Biol. Sport 2020, 37, 313–319.

Reviewer 2 Report

The manuscript titled "Pandemic within a pandemic: Increase in the number of injuries during and post Covid-19 lockdown." deals with a contemporary topic: the changes in individuals' life after the Covid-19 lockdown. The authors aimed to assess the onset of injuries during and after the lockdown, among 118 physically active young adults. The findings showed that 80% of athletes practiced aerobic training, 2/3 injury location was the lower limb and of all the injuries recorded 67 occurred during the first period of lockdown.

The manuscript presents several conditions that may comprise its publication. Generally, some sentences are asserted without a supportive citation and the statistical analysis performed is poor. Additional statistical methods are required to assert some findings.

Title

If the aim to call the paper "Pandemic within a pandemic" was to be somehow fun, I do not find it pleasant. This is not a pandemic inside a pandemic, it's just the analysis of injuries during the pandemic period. It is overly ambitious to call it pandemic.

Introduction

L37-38 This sentence does not introduce the contents of the manuscript correctly. The authors are referring to young adults, but this sentence refers to children, elders, and adults with chronic diseases. I would suggest revising it and focusing on the study's population.

L38-40 Are people still isolated at home is some countries? If yes, you should add a citation for it.

L43-46 This sentence is correct but needs a citation; you may read this article (doi: 10.3390/ijerph17186567)

L57-62 That general overview of the incidence of lockdown among different countries is interesting. I suggest you also read this paper (doi: 10.3390/ijerph18168680) so you can compare it with the German student population.

L75-80 Authors need a citation to say it.

Methods

L89-91 Authors could add the questionnaire link so readers may consult it.

L109 The p of p-value goes in lowercase, fix it.

Poor methods are applied to analyze the data. I would suggest using the Kendall–tau rank correlation coefficient to compare the variables and the Odds Ratio to evaluate the incidence of injuries.

Results

L122-26 The symbol of the Chi-square is χ² and not the normal X2, fix it.

L123 what does the number in brackets mean after the Chi-square symbol? E.g., X2 (16) (and also in the subsequent cases).

L124 is “positive and moderate correlation (CC = 0.69 p < 0.05).” Actually, the material and methods do not describe a statistical method to assess the correlation between variables. (As suggested for methods, use Kendall-tau).

Discussion

The discussion session is too long compared with the rest of the manuscript. I would suggest revising it and removing the sentence that the authors already discussed in the introduction.

L135 What is a "positive relationship"? If it is a statistical thought, authors cannot say it because no statistical methods to study a positive or negative correlation—eventually—are present.

L142-43 Even if these recommendations are widely known, the appropriate citation is required.

L156 what is a "stronger lower body"? stronger in terms of muscles? aerobic capacity?

L178-181 Authors never stated that their population is/is not an amateur population. Before asserting this sentence, you should clarify it.

Reviewer 3 Report

The main problem of this paper application of new questionnaire. Metric characteristics must be made before using a new measuring instrument. At least the reliability of the questionnaire should be reported. My suggestion is to calculate reliability if possible.

Round 2

Reviewer 2 Report

The authors correctly addressed the comments.

For the title, I like the new one proposed, however since the other reviewers did not comment it, feel free to change it or not. I believe that people might be more attracted by the second one.

After I checked your questionnaire, I agree that alternative statistical tests were not applicable. I never found researchers writing the degree of freedom after the chi-square results, but if you are familiar with it just go.

Furthermore,

- Authors should report in the tables the chi-square results. 

- In table 1, the authors divided the population by sex, but for the subsequent analysis, the population has been considered altogether; why?

Reviewer 3 Report

          You wrote me that this is not your questionnaire. Add a couple of information to the methods chapter, whose questionnaire it is, where it was used, how reliable it is.
